# Remediation of 1-Nitropyrene in Soil: A Comparative Study with Pyrene

**DOI:** 10.3390/ijerph17061914

**Published:** 2020-03-15

**Authors:** Shuo Li, Yatao Huang, Minhui Zhang, Yanchen Gao, Canping Pan, Kailin Deng, Bei Fan

**Affiliations:** 1Key Laboratory of Agro-products Quality and Safety Control in Storage and Transport Process, Ministry of Agriculture and Rural Affairs/Institute of Food Science and Technology, Chinese Academy of Agricultural Sciences, Beijing 100193, China; lishuo0610@126.com (S.L.); huangyatao@caas.cn (Y.H.); minhui0510@126.com (M.Z.); gaoyanchen1213@126.com (Y.G.); 2Department of Chemistry, College of Science, China Agricultural University, Beijing 100193, China; canpingp@cau.edu.cn

**Keywords:** *n*PAHs, 1-nitropyrene, pyrene, soil, remediation

## Abstract

Nitrated polycyclic aromatic hydrocarbons (*n*PAHs) are ubiquitous environmental pollutants, which exhibits higher toxicity than their corresponding parent PAHs (*p*PAHs). Recent studies demonstrated that the *n*PAHs could represent major soil pollution, however the remediation of *n*PAHs has been rarely reported. In this study, biological, physical, and chemical methods have been applied to remove 1-nitropyrene, the model *n*PAH, in contaminated soil. A comparative study with pyrene has also been investigated and evaluated. The results suggest that the physical method with activated carbon is an efficient and economical approach, removing 88.1% and 78.0% of 1-nitropyrene and pyrene respectively, within one day. The zero-valent ion has a similar removal performance on 1-nitropyrene (83.1%), converting 1-nitropyrene to 1-aminopyrene in soil via chemical reduction and decreasing the mutagenicity and carcinogenicity of 1-nitropyrene. Biological remediation that employs scallion as a plant model can reduce 55.0% of 1-nitropyrene in soil (from 39.6 to 17.8 μg/kg), while 77.9% of pyrene can be removed by plant. This indicates that *n*PAHs might be more persistent than corresponding *p*PAHs in soil. It is anticipated that this study could draw public awareness of nitro-derivatives of *p*PAHs and provide remediation technologies of carcinogenic *n*PAHs in soil.

## 1. Introduction

Polycyclic aromatic hydrocarbons (PAHs) are frequently identified as contaminants in the environment and food. Due to their potent mutagenicity, long-term effects, persistence, and bioaccumulation, a large number of studies have been undertaken to focus on protecting public health from *p*PAHs [1,2,3]. Nitrated polycyclic aromatic hydrocarbons (*n*PAHs) are nitro-derivatives of *p*PAHs, which can be generated by the same sources of *p*PAHs or by secondary reactions of *p*PAHs with NO_x_ [4,5]. Despite being present at lower concentrations than *p*PAHs, *n*PAHs are proven to be more carcinogenic and mutagenic to humans [6]. For example, 1-nitropyrene, the most abundant *n*PAHs emitted from diesel engines, has been classified by the International Agency for Research on Cancer as a probable carcinogen to humans. [7]. Toxicity data also demonstrated that 1-nitropyrene was over ten times more carcinogenic than pyrene and accounts for more than one-quarter of the mutagenicity of diesel emissions [6,8,9]. Due to their wide occurrence and potent toxicity, preventions from *n*PAHs and *p*PAHs are equivalently necessary.

Due to the hydrophobic properties of *p*PAHs and their derivatives, soil is the major compartment of *p*PAH pollution in the environment, and PAH-contaminated soils have been reported to contain significant amounts of other polycyclic aromatic compounds (PACs), such as alkylated PAHs, heterocyclic PACs (containing oxygen and sulfur), as well as more polar nitro-PAHs (*n*PAHs) [10]. Recently, emerging evidence reported the presence of *n*PAHs in soil [11,12,13,14,15]. Sun et al. monitored the nitro-PAHs in agricultural soils in Eastern China and found the total concentration of *n*PAHs was 50 μg/kg [12]. The *n*PAHs level of surface soil in the Yangtze River Delta was in the range of 0.4–4.6 μg/kg, while that of Xian was 118 μg/kg [13,14]. In the surface soil of Japan, the concentrations of *n*PAHs ranged from 0.08 to 15.8 μg/kg [15]. The *n*PAHs in soil could be translocated to plants and accumulated to the human diet or affect human health via soil ingestion directly, which increases the exposure risk to *n*PAHs [12,16,17]. Thus, the remediation of *n*PAHs in the soil is extremely needed. 

For more than a decade, environmental experts focused on developing several chemical, physical, biological, and thermal technologies to remediate organic pollutants like *p*PAHs in soil, decreasing the risk of ecological consequences [18,19,20,21]. Physical remediation that is based on the principle of adsorption is one of the most widely used methods, as polycyclic aromatic compounds (PACs) are prone to absorb solid media [22]. Sorption materials like activated carbon and biochar are popular because of their low cost and high efficiency. In previous studies, activated carbon has been applied to sequester organic pollutants, e.g., DDT, PAHs, and PCBs, in field soil and sediments [22]. However, the further treatment of polluted media remains to be another challenge. Chemical remediation using microscale or nanoscale zero-valent iron has also been reported as a promising way to reduce nitro aromatic compounds or PAHs [23,24]. Ming-Chin Chang et al. found the removal of pyrene in soil sample was feasible by nanoscale iron addition, diminishing 62% of pyrene within 60 min by 0.15 g/g soil under ambient conditions [23]. Lavin et al. proposed and evaluated the strategy of using zero-valent iron to diminish nitrobenzene, as it can reduce nitro-containing contaminants to amino products [24]. However, the feasibility of remediating *n*PAHs by Fe has never been investigated. Bioremediation, which is safe and environmentally friendly, has also gained wide approval among remediation technologies for *p*PAHs, however the low efficiency and more toxic by-products might limit its application [25,26].

Potential alternatives for *n*PAHs removal/degradation from contaminated soils have rarely been proposed. In 2016, Falciglia et al. [10] firstly applied bench-scale microwave heating treatment to remove *n*PAHs in soil and was compared with *p*PAHs. In their study, *n*PAHs are more difficult to remove from soil than *p*PAHs. After 10 min treatment by 440 W MW, only 20–40% removal of *n*PAHs was observed, while 70–100% of *p*PAHs can be removed. However, in the study of *n*PAH removal, the applications and mechanisms understanding of other typical technologies, like physical adsorption, chemical reduction, as well as biological remediation have received limited attention. Thus, the investigation of removal techniques for *n*PAH is urgent and essential, which could benefit future research in the *n*PAH removal strategy design. 

Based on the above considerations, the major objectives of this study are: (i) to assess the potentiality, features, and kinetics of commonly used physical, chemical, and biological treatments of model *n*PAH (1-nitropyrene) in contaminated soil employing bench-scale experiments; (ii) to compare the removal efficiency of 1-nitropyrene and its *p*PAH (pyrene), as well as to provide technical strategies for the in-situ or ex-situ soil remediation. It is anticipated that these fundamental data would provide a basis for *n*PAH remediation for science and technology, and thus decrease the risk of human exposure to *n*PAHs. 

## 2. Materials and Methods

### 2.1. Reagents and Chemicals

1-Nitropyrene, 1-aminopyrene, pyrene and activated carbon (4–12 mm, pH = 7.0, zeta potential = –16 mV, BET = 528 m^2^/g) were purchased from Sigma (St. Louis, MO). HPLC-grade acetonitrile and methanol were purchased from Merk (Darmstadt, Germany). Acetic acid, iron powder, analytical-grade acetonitrile, sodium chloride (NaCl), and anhydrous magnesium sulfate (anhydrous MgSO_4_) were obtained from Sinopharm Chemical Reagent Co., Ltd. (Beijing, China), and mPCF columns were purchased from Lvmian Technologies Inc. (Beijing, China). Ultra-pure water was obtained from Wahaha Group Co., Ltd. (Hangzhou, China).

Soil was collected in Yuanmingyuan West Road, Beijing. It was air-dried in a fume hood, and then ground and sieved through a 100 mesh (0.149 μm) sieve to remove the debris and stones. The soil was then sealed in a glass jar and used as needed. The physical and chemical properties of the experimental soil are shown in Table 1. 

### 2.2. Experimental Design

#### 2.2.1. Soil Contamination 

1-Nitropyrene or pyrene-contaminated soil samples (50 μg/kg) were prepared by spiking to topsoil samples with known amount of 1-nitropyrene or pyrene in acetone. Afterwards, the soils were homogenized by mesh and vapored in fume hood for overnight. Concentrations of 1-nitropyrene and pyrene were determined by ultraperformance liquid chromatography -fluorescence detector (UPLC-FLD). The initial concentrations of 1-nitropyrene and pyrene in contaminated soil are 39.6 ± 3.2, 42.6 ± 3.5 μg/kg, respectively.

#### 2.2.2. Physical Remediation by Activated Carbon 

Activated carbon (2.5 g) was added onto the 1N-pyr or pyr contaminated soil (10 g), after certain times (1, 2, 4, 7, 16 h), the activated carbon was filtered with a 100 mesh sieve to remove activated carbon before soil samples were analyzed for 1N-pyr or pyr using UPLC-FLD.

#### 2.2.3. Chemical Remediation by Zero-Valent Iron 

The chemical remediation of 1N-pyr and Pyr was conducted in a similar way. Iron powder (10 g) was added to the contaminated soil (10 g) to transform the nitro group to amino group, after 1, 2, 4, 7, and 16 h (*n* = 3), the iron powder was removed by magnet and soil samples were analyzed for 1N-pyr or pyr using UPLC-FLD. 

#### 2.2.4. Biological Remediation by Vegetation

The biological remediation was performed by cultivating scallion in 1-nitropyrene or pyrene-contaminated soil. The scallions (*n* = 36) used were purchased in a market in Haidian District Beijing. After cutting their leaves, scallions were planted in pots separately. The plants were left in the ambient environment with natural daylight and were irrigated once per week. After the plants were grown in the soil for 3, 7, 14, 21, 28, and 35 days (*n* = 3), the soil samples were separated and homogenized separately for UPLC-FLD analysis. A control experiment was conducted by analyzing soils without vegetation. 

### 2.3. pPAH and nPAH Analysis

#### 2.3.1. Extraction and Clean-up

Soil samples were homogenized and sieved by 0.2 mm mesh, and 10 mL of hexane was added to 10 g of soil samples, then the mixture was vortex for 1 min. NaCl (4 g) was then added to the samples and vortex for 1 min, then sonicate for 15 min. After centrifugation at 6000 rpm for 10 min, 2 mL of the supernatant was extracted and put into a multi-plug filtration cleanup (*m*PFC) column to clean-up. The supernatant was then transferred into a 1.5 mL microcentrifuge tube and dried under N_2_ stream for direct pyrene and 1-aminopyrne analysis or underwent Fe/H^+^ treatment for 1-nitropyrene analysis (Figure 1).

#### 2.3.2. Derivatization 

Using a previously reported method [27], samples for 1-nitropyrene analysis were treated with Fe/H^+^ to reduce the non-fluorescing 1-nitropyrene to fluorescing 1-aminopyrene for its indirect analysis by UPLC-FLD. In brief, the residue was re-dissolved in 200 µL of 15% acetic acid in methanol (v/v), and 10 mg iron powder was added. The sample mixtures were then vortexed vigorously for 20 min to reduce 1-nitropyrene to 1-aminopyrene before the supernatant was analyzed by UPLC-FLD.

#### 2.3.3. Instrumentation

UPLC-FLD analysis was performed on a Waters e2695 HPLC system (Palo Alto, Santa Clara, CA, USA) coupled to a programmable fluorescence detector (Waters 2475 FLR Detector). The sample extract was injected onto a Waters HSS T3 column (100 × 2.1 mm, 1.8 µm, 5 μL injection volume) at 40 °C for chromatographic separation. The column was eluted with a binary solvent system of water (A) and acetonitrile (B). Gradient elution at constant flow rate of 0.2 mL/min was used. The solvent gradient started from 30% B and was programmed to linearly increase to 100% B in 12 min, hold for another 5 min, and then recondition at 30% B. The eluate was monitored by FLD, which was time-programmed at excitation/emission wavelengths as follows: 0–10 min: λ_ex_ 240 and λ_em_ 435 nm (for 1-aminopyrene); 10–20 min: λ_ex_ 240 and λ_em_ 390 nm (for pyrene). As shown in Figure 2, the 1-aminopyrene eluted at 9.42 min, and pyrene eluted at 11.35 min.

### 2.4. Data Analysis

#### 2.4.1. 1-Nitropyrene Concentration

As the derivatization step, which converts 1-nitropyrene to 1-aminopyrene, was included before UPLC-FLD analysis, the initial concentration of 1-aminopyrene in soil might influence the results. Thus, the 1-aminopyrene concentrations before and after derivatization were detected. The concentration of 1-nitropyrene was calculated using Equation (1), where $\rho_{t}$ represents the concentration of total 1-aminopyrene after the Fe/H^+^- induced nitro reduction and $\rho_{0}$ is the native concentration of 1-aminopyrene obtained by analyzing the underivatized sample.
(1)$$\rho =\rho_{t}-\rho_{0}$$

#### 2.4.2. Removal Rate 

The removal percentage (R%) was calculated by Equation (2), where C_t_ and C_0_ represents the final and initial concentration of pollutant, respectively.
(2)$$\%R=\frac{C_{0}-C_{t}}{C_{0}}\times 100$$

#### 2.4.3. First order Kinetics

The dissipation of 1-nitropyrene and pyrene in soil were fitted using a method reported previously [28]. In brief, concentrations of 1-nitropyrene and pyrene in soil were plotted against the time of sampling. For both contaminants, a simple first-order degradation kinetic model (SFO; Equation (3)) was used due to the 50% decline.
(3)$$C_{t}=C_{0}e^{-kt}$$

In Equation (3), *C_t_* is the concentration of contaminants remaining in soil (μg/kg) after *t* days, *C_0_* is the initial concentration of contaminants (μg/kg), and *k* is the rate of degradation (day^−1^).

## 3. Results and Discussion

### 3.1. Analytical Method Validation

The method performance was evaluated by recovery (accuracy) and repeatability (precision), which were studied by spiking 1N-pyr or pyr in soil matrix at three concentration levels (0.5–50 μg/kg). Average recoveries of 1-nitropyrene and pyrene in soil samples were evaluated and calculated using matrix-matched calibration standards. The results (Table 2) showed that good recoveries (86.8%~104.7%) were obtained for both contaminants in soil matrices. Relative standard deviations (RSDs) of the intra-day reproducibility were under 8.9% (*n* = 5) of the analytical method for 1-nitropyrene and pyrene analysis. Over a period of one month, the RSDs for the studies ranged from 3.7%–10.2% (*n* = 5). The data on method accuracy and precision indicated that the *m*PFC based method met the requirements for analyzing 1-nitropyrene and pyrene in soil samples.

### 3.2. Physical Remediation by Activated Carbon 

Sorption has been found to be a promising technique to remove PAHs due to its low cost, simple operation, and less by-products formation [21]. Among the sorption media, activated carbon was a popular one to remove PAHs because of its large surface area [22,29]. In this study, the activated carbon could diminish 88.1% of 1N-pyr and 78.0% of pyr after 16 h in contaminated soil (Table 3), and the half-lives of 1N-pyr and pyr dissipations were 0.8 and 1.5 days (Figure 3A), respectively. The Van der Waals forces may explain the removal rate discrepancy between 1N-pyr and pyr. As Van der Waals forces increase with molecular weight, the adsorption between 1N-pyr and activated carbon was stronger than that of pyr. At the same time, the N and O in the nitro group of 1N-pyr might also form hydrogen bonds with the suspended hydrogen atoms on activated carbon as this commercial material was activated by acid [30].

Based on the results, the remediation by activated carbon showed a fast rate, which removed around 80% of the pollutant within four and seven hours for 1N-pyr and pyr, respectively, indicating the physical remediation is an efficient method. However, the activated carbon presented similar absorption characteristics of the pollutants without selectivity, in which the absorption ability may be weakened when the soil is collected from a contaminated source.

### 3.3. Chemical Remediation by Zero-Valent Iron 

Chemical remediation, which relies on the chemical redox, exhibits better selectivity than the sorption approach. Among chemical reagents, zero-valent iron has been found to be a superior reductant because it is relatively inexpensive, abundant, harmless to the environment, and effective in reducing organic contaminants [24]. Under the acidic condition, zero-valent iron could accelerate the conversion of nitro- group to amino group to a non-toxic amino group to reduce the overall toxicity of the soil. Based on this principle, 10 g of iron powder was added to 10 g of contaminated soil (pH = 5.4), and after 1, 2, 4, 7, 16 h treatment, the 1-nitropyrene and pyrene residue was analyzed. According to the monitoring, the pyrene cannot be removed by the addition of Fe as expected due to the chemical stability of the aromatic structure. As a contrast, over 83% of 1-nitropyrene was removed within 16 h, which is competitive with activated carbon. Table 4 demonstrates the removal effects of 1-nitropyrene using activated carbon and zero-valent iron; activated carbon could absorb most of the 1-nitropyrene after 4 h, while the zero-valent iron needed 7 h to achieve the plateau (Figure 3B). However, when we changed the acidic soil to alkaline soil, the 1-nitropyrene in the soil cannot be removed.

Compared with the activated carbon, the chemical remediation by zero-valent iron has better selectively to the pollutant, however, at the same time, the chemical remediation needs strict control of the soil (reaction) condition. For example, when the source condition is changed from acidic soil to alkaline soil, the zero-valent iron loses the reaction activity, and the 1-nitropyrene in the soil cannot be reduced and remains in the soil as a pollutant.

### 3.4. Biological Remediation by Scallions

Except for the physical and chemical remediation methods, another efficient and eco-friendly method is biological remediation, which has gained wide approval among remediation technologies [25,26]. In this study, scallion was used for biological remediation because it has a deep root and requires less maintenance. From the result, scallion successfully removed 55.0% of 1-NP and 77.9% of pyr in the contaminated soil after 35 days, as shown in Figure 3C. Compared with the previous methods, the biological remediation needs a longer period for the pollutants removal. At the same time, the removal rate of 1N-pyr and pyr also indicated that the biological remediation had the target selectivity.

Figure 3 plotted the concentration versus time of each remediation method and was fitted by the first-order kinetic decay model. Table 5 summarizes the plant-assisted dissipation rate constants (*k*) and half-lives (*T_1/2_*) of 1-nitropyrene and pyrene in agricultural soil in the presence of agents. For the biological remediation, the dissipations of 1-nitropyrene and pyrene fitted well (R^2^, 0.9960–0.9973) with the first-order kinetic decay model (Equation (2)), where the kinetic constants and half-lives could also be obtained (Figure 3C). The first-order model revealed that pyrene degraded more rapidly and completely than 1-nitropyrene with vegetation, which was consistent with previous studies on phenolic compounds, suggesting that the addition of a nitro group to the ring markedly enhanced its resistance to soil degradation [31,32]. In general, based on half-life (*T*_1/2_) values, approximately 23.1 and 11.6 days are needed for degrading 50% of 1-nitropyrene and pyrene in soil, respectively. According to previous literature, the plant could accumulate/sequester/chemically transform the contaminants, and/or manipulate the microenvironment of soil [33]. The root exudate stimulated by 1-nitropyrene/pyrene could mediate the microbial community in the rhizosphere soil, which may promote bacterial activities and degrade pollutants [34]. 

A challenging problem for bioremediation would be that more toxic metabolites or by-products of original contaminants may be produced in the degradation process. For example, Bandowe et al. found that specific PAHs could transform into oxy-PAHs during remediation, which were more carcinogenic [11]. In terms of 1-nitropyrene, the major degraded product of 1-NPyr was 1,6-pyrenedione and 1,8-pyrenedione [35], both of which are non-carcinogens suggested by IARC [36], resulting in lower risks of bioremediation methods.

In contrast, the chemical and physical remediation methods show a rapid pollutant removal rate, which is two order magnitude higher than that of biological remediation. However, considering the application scenes, the biological remediation method would attract broader interests for pollutant removal, not only for the laboratory use, but can also be used for the soil clean of greenhouse and field soil. Meanwhile, for the chemical and physical remediation methods, after the purification of plant soil, the removal of remediation agent is also needed, which indeed limits the real application to laboratory use and greenhouse. 

### 3.5. Comparison of Remediation Technologies

According to the results, the physical and chemical remediation methods used in this study showed competitive efficiency of removing 1-nitropyrene in soil, both of which decreased more than 80% of 1-nitropyrene after 16 h. The physical remediation by activated carbon showed poor selectivity of the pollutants, while chemical remediation by zero-valent iron could only transform 1-nitropyrene. However, the chemical reduction required more severe conditions compared with sorption methods. 

The bioremediation with scallions seemed to be less effective for nitro-PAHs, as merely 55.0% of 1-nitropyrene was degraded after 35 days, and pyrene was degraded more rapidly than its nitro-derivative. However, considering the remediation of in-situ soil, the biological remediation avoids transferring the pollutants to another media, while the secondary pollution from the physical and chemical remediation agents needs to be highly avoided and controlled. 

Thermal treatment optimized by Falciglia was also efficient for *n*PAHs, removing about 90% after 60 min irradiation (440 W), but the application scene is limited for ex-situ dry soils [10].

Overall, compared with PAHs, the remediation of *n*PAHs are still at the prototype level. More studies and field applications are needed to achieve risk-based and green remediation of *n*PAHs.

## 4. Conclusions

The main conclusions for this study are:

The physical remediation utilizing activated carbon to remove 1-nitropyrene and pyrene was effective and economical, which degraded more than 80% of pollutants within 7 h.

Chemical technology that uses zero-valent iron demonstrated a similar ability to remove 1-nitropyrene (83.0%) with better selectivity compared with activated carbon. However, at the same time, the chemical remediation needs strict control of the soil (reaction) condition.

Bioremediation, especially the phytoremediation, was time-consuming but eco-friendly, which dissipated 55% of 1-nitropyrene and 77.9% of pyrene after the growth period. However, considering the remediation of in-situ soil, the biological remediation avoids transferring the pollutants to another media, while the secondary pollution from the physical and chemical remediation agents needs to be controlled. 

It is anticipated that this study could draw public awareness of nitro-derivatives of *p*PAHs and provide remediation technologies of carcinogenic *n*PAHs in soil.

## Figures and Tables

**Figure 1 ijerph-17-01914-f001:**
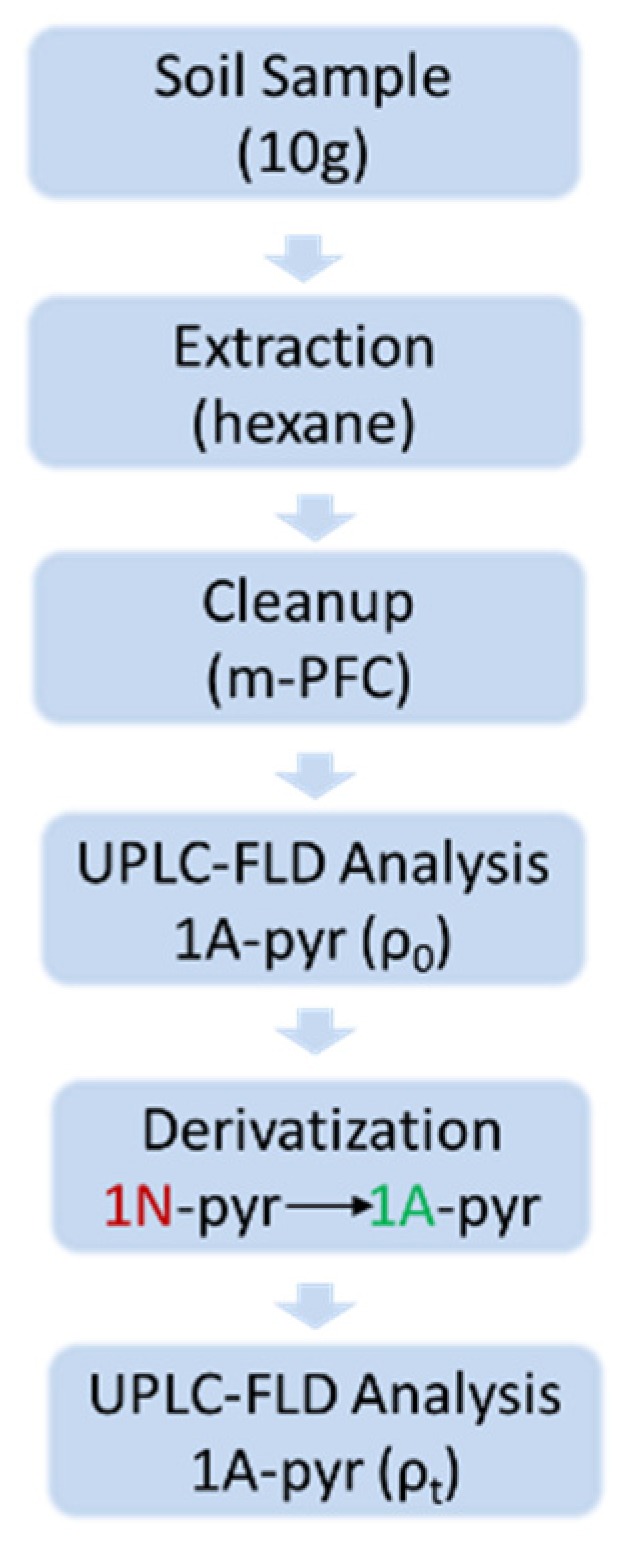
Analysis procedure of 1N-Pyr. The concentration of 1N-pyr was calculated by Equation (1).

**Figure 2 ijerph-17-01914-f002:**
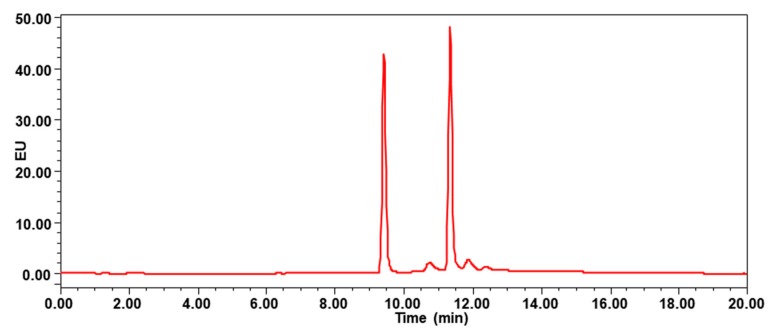
Chromatogram of mixture standard at 20 μg/L, the 1-aminopyrene eluted at 9.42 min, and pyrene eluted at 11.35 min.

**Figure 3 ijerph-17-01914-f003:**
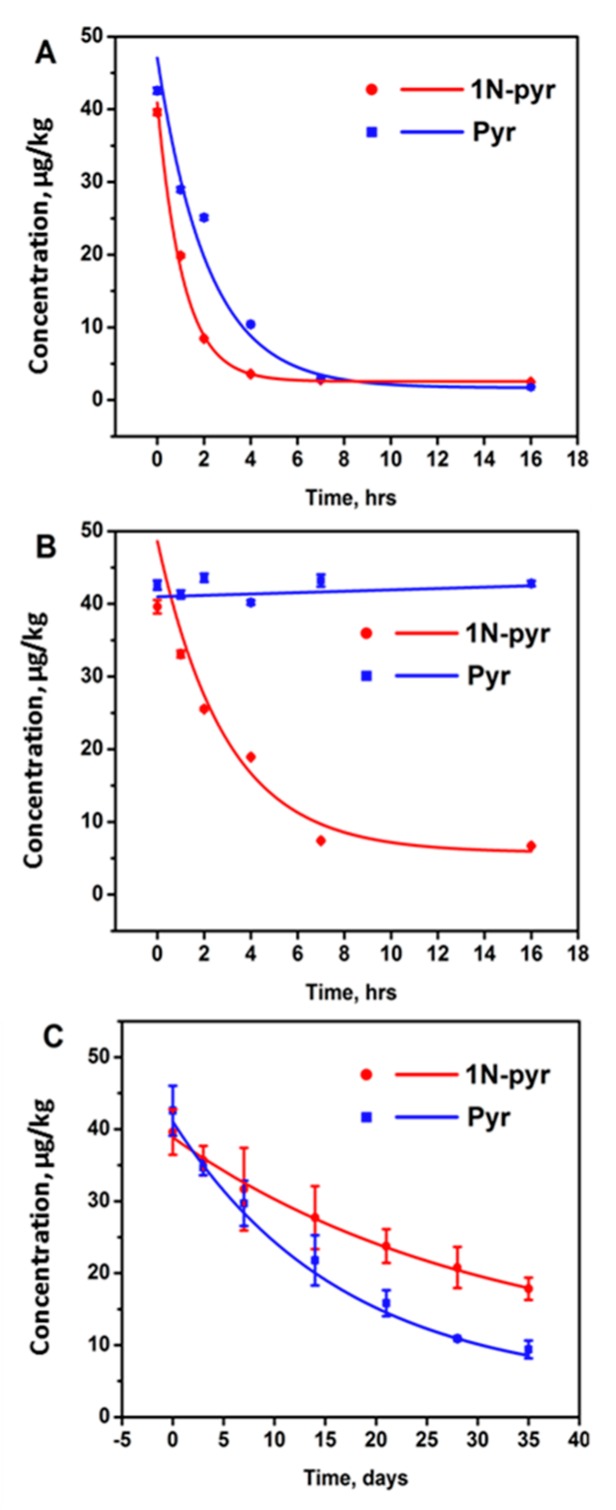
Dissipation of 1-nitropyrene and pyrene in soil by activated carbon (**A**), zero-valent iron (**B**), and scallion (**C**).

**Table 1 ijerph-17-01914-t001:** Physiochemical properties of experimental soil.

Property	Value ^a^	Methods
pH	8.82 ± 0.04	NY/T 1121.2–2006
Organic carbon(g/kg)	3.33 ± 0.15	LY/T 1237–1999
Cation exchange capacity cmol(+)/kg	10.01 ± 0.32	LY/T 1243–1999

^a^ The values were determined for three times.

**Table 2 ijerph-17-01914-t002:** Validation parameters of analytical method.

Comp. and Conc.		Accuracy		Precision	
	Spiked Conc.	Calculated Conc.	Recovery	Intra-Day	Inter-Day
	(μg/kg)	(μg/kg)	(%)	(% RSD)	(% RSD)
	0.5	0.48 ± 0.04	95.9 ± 8.4	7.6	6.7
1N-pyr	5	5.24 ± 0.48	104.7 ± 9.1	8.9	10.2
	50	43.40 ± 2.04	86.8 ± 4.7	3.8	6.6
	1	1.08 ± 0.08	108.0 ± 7.1	5.4	10.0
Pyr	5	4.54 ± 0.30	90.8 ± 6.5	1.7	3.7
	50	47.85 ± 4.55	95.7 ± 9.5	2.2	4.3

RSD: Relative standard deviation.

**Table 3 ijerph-17-01914-t003:** Removal effects of contaminants in soil by activated carbon.

Time (h)	1N-Pyr	Pyr
0	100.0 ± 1.2%	100.0 ± 0.7%
1	55.7 ± 0.3%	87.0 ± 1.9%
2	26.7 ± 0.2%	46.4 ± 0.2%
4	12.9 ± 0.1%	28.2 ± 1.1%
7	11.7 ± 0.1%	20.3 ± 0.8%
16	11.9 ± 0.0%	22.0 ± 0.3%

**Table 4 ijerph-17-01914-t004:** Comparison of removal effects of 1-nitropyrene using activated carbon and zero-valent iron under acidic soil.

Time (h)	Activated Carbon	Zero-Valent Iron
0	100.0 ± 1.2%	100.0 ± 1.4%
1	55.7 ± 0.3%	83.6 ± 2.1%
2	26.7 ± 0.2%	64.5 ± 0.9%
4	12.9 ± 0.1%	47.8 ± 1.2%
7	11.7 ± 0.1%	18.7 ± 0.8%
16	11.9 ± 0.0%	16.9 ± 0.2%

**Table 5 ijerph-17-01914-t005:** First-order rate constants (*k*) and half-lives (*T_1/2_*) of 1-nitropyrene and pyrene degradation in contaminated soils with scallions.

Parameters	Activated Carbon	
	**1N-pyr**	**Pyr**
Regression equation	C_t_ = 38.34e^−0.90t^	C_t_ = 45.38e^−0.46t^
Determinant Coefficient (*R^2^*)	0.9880	0.9427
Rate constant (*k*)	0.90	0.46
Half-life (*T_1/2_*_,_ day^−1^)	0.8	1.5
Removal rate (%)	88.1	78.0
	Zero-valent iron	
	1N-pyr	Pyr
Regression equation	C_t_ = 42.84e^−0.34t^	
Determinant Coefficient (*R^2^*)	0.8578	
Rate constant (*k*)	0.34	
Half-life (*T_1/2_*_,_ day^−1^)	2.0
Removal rate (%)	83.0	
	Scallion	
	1N-pyr	Pyr
Regression equation	C_t_ = 30.6e^−0.03t^	C_t_ = 37.1e^−0.06t^
Determinant Coefficient (*R^2^*)	0.9973	0.9960
Rate constant (*k*)	0.03	0.06
Half-life (*T_1/2_*_,_ day^−1^)	23.1	11.6
Removal rate (%)	55.0	77.9

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
