# Peer review of "Remediation of 1-Nitropyrene in Soil: A Comparative Study with Pyrene"

_ijerph, 2020, doi:10.3390/ijerph17061914_

Round 1

Reviewer 1 Report

English should be improved, especially in the new added sections.

Author Response

We express our sincere gratitude to the reviewer for the invaluable suggestions to improve the quality of our manuscript. Line 220-222 and line 242-244 (highlighted by yellow color) have been revised according to comments. Also, we have gone through the manuscript again carefully, and the errors in grammar and wording are now corrected in the revised manuscript. (highlighted by red color).

Reviewer 2 Report

It can be accepted as the present form.

Author Response

We express our great appreciation to the reviewer for comments on our paper.

This manuscript is a resubmission of an earlier submission. The following is a list of the peer review reports and author responses from that submission.

Round 1

Reviewer 1 Report

This paper compares various methods to remediate the by-product of combustion emitted from the diesel engines (1-Nitropyrene). Generally PAH soil contamination is a very relevant global problem. N-PAHs are more carcinogenic than PAHs, having higher toxicological significance at lower concentrations. Despite of this fact not so many removal possibilities have been studied. The idea for the paper is interesting. However the form in which it is presented reminds me more of a technical report than a scientific paper. The discussion and introduction is very poor, not enough literature research is done to support the results. The authors describe the results that clearly indicate the differences between the methods, but very less "science behind" is done. How did they got the idea? Were the methods used for other contaminants? if yes what were the results? Why are the results different? Authors have to work on their paper to elaborate the science hidden behind their results. If necessary additional analysis is recommended, as well as the language correction.

Reviewer 2 Report

The authors used the batch and greenhouse experiments to examine the remediation effect of nitrated polycyclic aromatic hydrocarbons (nPAHs) in soil by activated carbon, iron powder and plant. The topic is interesting, therefore, the manuscript should interest wider community of researchers who are working on the environmental fate of nPAHs contaminated soil in situ treatment. However, the manuscript needs major revisions and state the more detail of materials and methods before publication. Without a clear resolution of this issue I cannot give my positive recommendation. The major points that should be addressed by the authors are as follows:

Specific comments:

A very brief introduction, without providing any hypothesis. L70-72: Please provide the basic physical and chemical properties of the experimental soil. In similar, the concentration of 1-nitropyrene and pyrene before the batch test should be added in the manuscript. The data of pH, zeta potential, BET surface area of activated carbon should be added? Line 85: I'm confused how to filter activated carbon from contaminated soil? Please show pre-treatment such as filtration before soil sample analysis. Line 90-91:the authors stated that the iron powder was removed by magnet. Actually, there are also a lot of magnetic substances in the soil. Did magnet also attract other magnetic substances during the process of iron powder adsorption? I think it is not just iron powder. In section 2.2.4, the biological remediation by vegetation, plant samples collected, measured and recover rate of pollutants should be added in the manuscript. Line 131-132: this sentence “where pt represents the concentration of total 1-aminopyrene after the Fe/H+ induced nitro reduction”. It seems to be not related with the purpose of this study. At least, I cannot understand the connection of this sentence to the purpose of this study. I suggested all values of the manuscript are presented as mean ± standard error (n = 3), or all values are presented as mean ± standard deviation (n = 3). Very poor discussion, mostly repeating the results section.
